# Matrix Tablets Based on Chitosan–Carrageenan Polyelectrolyte Complex: Unique Matrices for Drug Targeting in the Intestine

**DOI:** 10.3390/ph15080980

**Published:** 2022-08-09

**Authors:** Alena Komersová, Roman Svoboda, Barbora Skalická, Martin Bartoš, Eva Šnejdrová, Jitka Mužíková, Kevin Matzick

**Affiliations:** 1Department of Physical Chemistry, Faculty of Chemical Technology, University of Pardubice, Studentská 95, 532 10 Pardubice, Czech Republic; 2Department of Analytical Chemistry, Faculty of Chemical Technology, University of Pardubice, Studentská 95, 532 10 Pardubice, Czech Republic; 3Department of Pharmaceutical Technology, Faculty of Pharmacy in Hradec Králové, Charles University, Akademika Heyrovského 1203, 500 05 Hradec Králové, Czech Republic

**Keywords:** polyelectrolyte complex, carrageenan, chitosan, drug targeting

## Abstract

The present study focused on the more detailed characterization of chitosan–carrageenan-based matrix tablets with respect to their potential utilization for drug targeting in the intestine. The study systematically dealt with the particular stages of the dissolution process, as well as with different views of the physico-chemical processes involved in these stages. The initial swelling of the tablets in the acidic medium based on the combined microscopy–calorimetry point of view, the pH-induced differences in the erosion and swelling of the tested tablets, and the morphological characterization of the tablets are discussed. The dissolution kinetics correlated with the rheological properties and mucoadhesive behavior of the tablets are also reported, and, correspondingly, the formulations with suitable properties were identified. It was confirmed that the formation of the chitosan–carrageenan polyelectrolyte complex may be an elegant and beneficial alternative solution for the drug targeting to the intestine by the matrix tablet.

## 1. Introduction

Due to the specific physico-chemical properties and the advantages such as biocompatibility, biodegradability, and low toxicity, some types of natural polymers (e.g., polysaccharides) are increasingly used in various oral dosage forms. A combination of biopolymers of a different nature allows researchers to modify a drug release mechanism and achieve a desirable drug release profile. Nowadays, the challenge lies in the design and usage of a combination of polysaccharides of anionic and cationic natures for the formulation of extended oral dosage forms. Study of these biopolymers combinations can significantly widen the scope of the design of matrix systems.

Chitosan (CHTS), a copolymer of N-acetyl-D-glucosamine and D-glucosamine, is obtained via an alkaline deacetylation of chitin [1,2]. The physico-chemical and biological properties of CHTS (solubility, chemical reactivity, and antimicrobial and antioxidant activity), as well as its medical and pharmaceutical applications, are affected by the degree of deacetylation, which can range from 40 to 90%. CHTSs with a low degree of deacetylation (≤40%) are soluble in pH up to 9, while highly deacetylated CHTSs (≥85%) are soluble only in acidic solutions (pH up to 6.5). Due to a charge repulsion, the molecules of highly deacetylated CHTS form extended structures with more flexible chains compared to CHTS with a lower deacetylation degree. This can explain the fact that viscosity of aqueous solutions of CHTSs increases with an increasing deacetylation degree. CHTS is a weak base (with a p*K*a value in the range of 6.2–7.0). In an acidic medium, the amino groups of CHTS are protonated, resulting in a positively charged biopolymer that can interact with the negatively charged molecules [3,4,5].

Carrageenans (CRGs) belong to the group of natural polysaccharides that are extracted from red algae. They consist of 1,3-linked β-galactose (G-units) and 1,4-linked α-galactose (D-units). The galactose units can be partially substituted by sulfate groups and therefore CRGs are biopolymers of anionic nature. The CRGs can be classified based on the degree of groups substitution. Three major types are κ, ι and λ-CRGs [6,7]. The κ-CRGs have only one sulfate group, ι-CRGs have two and λ-CRGs have three sulfate groups per disaccharide unit. These structural differences, together with the presence of 3,6-anhydro bridges (in the structure of κ-and ι-CRGs), influence the physico-chemical properties of CRGs, especially their linear charge density [8] and solubility [9,10]. The key property of CRGs for their use in drug-delivery systems is their swelling and ability to form a gel layer. Such gel-formation properties are characteristic of κ- and ι-CRGs. The presence of anhydro bridges is probably the key factor in gel-structure formation. λ-CRGs do not contain anhydro bridges and thus do not form gels [8]. Gel formation is a two-step process controlled particularly by temperature and the presence of cations (K^+^ or Ca^2+^). The process includes a transition from a random coil to a helix structure and then an aggregation of helices [6]. κ- and ι-CRGs form strong, rigid, and brittle gels with the cations mentioned above. κ-CRGs also show gel-formation properties under salt-free conditions [7]. The formation of a gel structure by CRGs is a thermally reversible process [9].

An electrostatic interaction between positively charged CHTS and negatively charged CRG results in pH-sensitive polyelectrolyte complex [6], which is a suitable tool for the formulation of matrix tablets with a controlled drug release.

A system of cationic–anionic polymers can be utilized in two different ways [11]: (1) the preparation of a polyelectrolyte complex (PEC) via a chemical reaction in the solution, then drying, crushing and sifting it to a desired particle size, and then use of the PEC as a matrix for the controlled drug release [12,13,14]; (2) the simple use of physical mixture of cationic and anionic polymers for the preparation of matrix tablets using a direct compression method and subsequent in situ formation of PEC [15,16,17,18,19,20]. The physical-mixture-based matrix tablets can be prepared more easily compared to the corresponding PEC prepared using a chemical reaction, and some experimental results confirmed that they have an excellent capacity for controlled drug release [21].

Although CHTS/CRG-based matrix tablets have been intensively studied in recent years, a number of questions remain unanswered. The present study focused on the more detailed characterization of these systems with respect to their potential application for drug targeting in the intestine.

## 2. Results and Discussion

The present section will systematically deal with the particular stages of the dissolution process, as well as with different views on the physico-chemical processes involved in these stages. In accordance, the results and the corresponding discussion will be split into the following sub-sections. In the *first sub-section*, the initial swelling of the tablets (primarily immersed in the acidic medium) will be reported based on the combined microscopy–calorimetry point of view. The *second sub-section* will focus on the pH-induced differences in the erosion and swelling of the tested tablets, as well as on the morphological characterization of the tablets. The *third sub-section* will report on the dissolution kinetics, as studied by means of the in vitro dissolution testing of the drug releases. In the *fourth sub-section*, the dissolution kinetics will be correlated with the rheological properties and mucoadhesive behavior of the tablets and, correspondingly, the particularly beneficial formulations will be identified.

### 2.1. Initial Swelling of Tablets in the Acidic Medium

The initial swelling of the tablets immersed in the acidic medium was determined by the rate of medium penetration into the tablets. This penetration reflected the bonding conditions between the compounds contained in the formulations. In the present study, two model drugs were used in the formulations—SA as an example of a (water-)insoluble compound, and TH as an example of a water-soluble model drug. In order to reveal the possible identifying characteristics and interactions between the considered compounds contained in the matrix tablets, simple DSC heating scans were performed (at 30 °C·min^−1^) for each particular model drug and PEC-forming polymer (the corresponding DSC curves are displayed in Figure 1A). As seen in Figure 1A, the two model drugs (SA and TH) exhibited a single endothermic peak corresponding to their melting with respective onsets of ~159 and 181 °C. Carrageenan showed sharp exothermic decomposition with the onset at ~200 °C. Note that the complexity and position of the CRG decomposition peak could be mildly affected by the type of the DSC pan (open, pin-hole, or hermetically closed) and applied atmosphere (air/N_2_ and flow rate) in the present study, all measurements were performed under identical conditions to eliminate these effects. The complex nature of the CRG decomposition was in good correspondence with literature data [22,23,24,25,26], corresponding to the loss in the −OSO^3−^ groups from the pendant chains attached to the polymeric backbone and/or to the carbohydrate backbone fragmentation. Decomposition of the other biopolymer, CHTS, manifested itself as a slow exothermic burn above ~230 °C. As the primary goal of the present study was to explore the PEC formation between the CRG and CHTS, we first confirmed that DSC was capable of distinguishing the formation of such interaction (an idea akin to that published in [11]). In Figure 1B, the DSC signals for the dry binary tablet (containing only CRG and CHTS at a 1:1 ratio) and the same binary tablet subjected to the dissolution in the acidic medium for 1 h are displayed together with the DSC signals corresponding to the pure substances. The DSC curve for the dry binary tablet was essentially an overlap of the two signals corresponding to the pure CRG and CHTS, indicating no significant interaction at this stage (note that the CRG decomposition had a similar onset but was slower, prolonged by the higher temperatures due to the dilution by CHTS). On the other hand, the immersion in the acidic medium resulted in a complete disappearance of the CRG decomposition peak, indicating the interaction between CRG and CHTS, which prevented the decomposition of CRG. This confirmed the original premise and allowed for continuation of the research with fully loaded matrix tablets containing the model drugs and additional excipients.

During the initial screening, the model drug solubility was found to have a crucial impact on the swelling kinetics of the tablets. The tablets containing TH exhibited a high tendency toward disintegration as well as rapid liquid uptake, as the tablets were completely swollen within a few minutes of immersion in the acidic medium. Figure 1C shows the DSC curves for the TH-containing matrix tablets (dry and subjected to the acidic dissolution for 5 min). Whereas the dry tablet showed the melting of TH and decomposition of CRG, even the very short dissolution indeed led to the disappearance of the CRG decomposition peak (due to the probable similar CRG/CHTS interaction), as well as to apparent shift of the TH melting to the lower temperatures (which will be discussed later in Section 2.3). On the other hand, the tablets containing SA swelled gradually over the duration of the acidic dissolution phase. For this reason, the detailed investigation of the initial swelling kinetics was performed only for the matrix tablets containing SA.

The time dependence of the tablet swelling was checked using optical microscopy for the series of dissolution experiments (matrix tablets immersed in the tempered and stirred acidic medium), for which the tablets were taken out of the medium at pre-selected times (1, 5, 30, 60, and 120 min)—see Section 3.2.3 for more details. For several chosen formulations, more frequent sampling was done to verify the overall trends. The tablets in different stages of the dissolution process were photographed immediately after withdrawing from the acidic solution—first from the top view and then on the cross-section (with the cut taken vertically through the center of the tablet). Figure 2 shows examples of the cross-section micrographs taken during the dissolution series performed for the formulation F13 (CRG/CHTS = 3:1; Prosolv^®^ SMCC 90 used as the dry binder). It is apparent that the tablets swelled gradually (by absorbing the acidic medium), forming an increasingly thick wet layer as the medium penetrated into the tablet volume. The swelling proceeded equally from all directions (as evidenced by the uniformly thick wet layer), resulting in a shape roughly proportionate to the original tablets’ dimensions.

In order to quantify the swelling kinetics, the following quantities were determined for each combination of tablet formulation and dissolution time: tablet height *h* (measured in the cross-section view), vertical thickness of the dry tablet core *h*_dry_ (measured in the cross-section view), and tablet diameter *d* (measured in the top view). Evolution of these quantities with dissolution time is shown for the particular formulations depicted in Figure 3A–C. Rapidity of the overall swelling (expressed by *h* and *d*) seemed to be universally driven by the CRG/CHTS ratio; the more CRG, the faster the swelling. This may be explained by the presence of the –OSO_3_^−^ groups of CRG [2], which facilitated easier absorption of the polar medium (compared to the –NH_2_ groups of CHTS). The rapidity of the swelling was further supported by the presence of the CaSO_4_·2H_2_O (ionic salt) used as a dry binder. Interestingly, the final *h* and *d* values representing the magnitude of the swollen tablets seemed to show anisotropic behavior. The overall tablet height (thickness) was clearly driven by the type of dry binder used (see Figure 3A), where Prosolv^®^ SMCC 90 caused additional tablet expansion (compared to the alternative increased CRG and CHTS contents denoted as “no binder”—see Table 1), while CaSO_4_·2H_2_O led to the lowest overall *h* due to the very limited additional liquid uptake by this salt, as well as by its generally low solubility. On the other hand, in the case of the tablet diameter, no such clear dependence could be recognized. The final value of the tablet diameter (see Figure 3C) appeared to be primarily driven by the CRG content, as determined by the CRG/CHTS ratio and its additional increase in the “no binder” formulations.

Interesting conclusions also arose from Figure 3B, in which the rate of medium penetration into the tablet was quantified via the *h*_dry_/*h* ratio. Here, the initial penetration was clearly accelerated by the presence of the dry binder in the following order: CaSO_4_·2H_2_O (as expected from the ionic salt) > Prosolv^®^ SMCC 90 > no binder. However, this influence was overtaken after 20–30 min by that of the increasing CRG/CHTS ratio, which further dominated in the formulations containing the dry binder. Interestingly, a reverse trend in the medium penetration rate was found for the formulations with no binder, where the increasing CRG/CHTS ratio slowed down the actual medium transport into the tablet core (due to the formation of the gel interface, which was not disrupted by the presence of the dry binder). This effect was even more apparent for the sole *h*_dry_ quantity, as shown in Appendix A. It was only the formulations with no binder that that still had a partially dry tablet core at the end of the acidic dissolution phase. Note that here, the term “dry” is used from the visual point of view (distinguished by color, with clear borderline between the core and the outer gel layer). As will be argued in the following paragraphs, even this core had to be wetted to certain extent, although no gel formation took place.

In order to further quantify the CRG behavior and the rate of its interaction with CHTS transformation in the tablets, DSC heating scan experiments were performed using the samples taken from the specific locations of each tablet subjected to the partial dissolutions in the acidic medium (see Section 3.2.7). A typical example of the DSC curves corresponding to the samples taken from the “dry” tablet core is shown in Figure 3D—the data correspond to the formulation F15. The curves show an endothermic peak (at ~150 °C) indicating the melting of SA, followed by a complex exothermic effect (at ~200 °C) that corresponded to the decomposition of CRG, as discussed in detail in Figure 1. The decreasing exothermic signal of CRG then indicated the degree of its interaction with CHTS prior to the DSC experiment (i.e., during the acidic dissolution), which we utilized for the quantification of the spatial CRG reaction kinetics within the particular formulations. Before the actual presentation of the kinetics, three points need to be made: (1) the edges and sides of the tablets always immediately showed a complete disappearance of the CRG decomposition signal on the DSC curves, indicating its practically immediate interaction with CHTS after wetting with the acidic medium; (2) a certain portion of the acidic medium had to permeate the “dry” tablet cores, as evidenced by the gradual decrease of the CRG DSC signal, even if the samples were taken from the seemingly “dry” tablet cores; and (3) dissociation of CRG in the acidic medium was conditioned by the presence of either CHTS or SA, as the sole pure CRG immersed in the acidic medium still exhibited full decomposition (based on the study of the binary CRG/CHTS tablets (see Figure 1B), the interaction with CRG was certain, but that with SA cannot be ruled out yet). Based on the above-mentioned points, we wanted to demonstrate that the main cause for the disappearance of the DSC CRG peak was also the interaction between CRG and CHTS in the fully loaded matrix tablets. Practically only these two substances were present in all the tested formulations (a similar decrease, although very rapid, also was observed for the TH-based formulations), which left either their mutual interaction or the hydrolysis of CRG caused by HCl as the plausible causes of the vanishing of the CRG decomposition peak. The available literature [27,28,29] reported the acidic hydrolysis of CRG to be quite slow (>5 h for a considerable decline in CRG content), which we verified by exposing the pure CRG to HCl for 30 min, after which the DSC signal of CRG remained unchanged and equal to that of the dry CRG. The existence of the interaction between CRG and CHTS was further supported by the fact that the CRG DSC signal was largely affected even at extremely low levels of tablet core wetting, which indicated that the liquid medium played only a supporting/interconnecting role for the interaction to proceed. By immersing the tested matrix tablets in distilled water, we further disproved the need for the presence of H^+^ ions for the CRG/CHTS interaction to take place (the CRG DSC signal vanished from the DSC curve during the immersion in H_2_O, which was qualitatively similar to that during the proper dissolution test in the acidic medium). This indicated that the CRG/CHTS interaction was chemically independent from the surroundings, and the liquid medium only facilitated the contact between the two polymers. However, the immersion in water resulted in a significantly slower decrease in the CRG DSC peak compared to the contact with the acidic medium. This indicated that the interaction between the two polymers was pH-driven, and the initiation of the PEC formation had occurred already (and primarily) during the acidic phase.

The kinetics of CRG interaction with CHTS in the tablet cores is shown in Figure 3E,F (Figure 3F is a zoomed-in view of the graph in Figure 3E). The quantification of the decrease in the CRG DSC peak was expressed as the ratio of the enthalpies corresponding to the CRG decomposition and SA melting—assuming that the release of SA was approximately constant for all studied formulations (exceptions to this behavior will be mentioned and discussed below). As will be demonstrated in the following Section 2.3, the release of SA from the majority of the explored formulations was ~25% at the 120 min mark. However, the formulations containing the CaSO_4_·2H_2_O dry binder exhibited an SA release of only ~5% over the 120 min of the acidic dissolution, meaning that the corresponding Δ*H*_CRG_/Δ*H*_SA_ ratios depicted in Figure 3E,F decreased by a factor of ~1.5. On the other hand, the formulation nos. 5 (CRG/CHTS = 1:1; no binder) and 7 (CRG/CHTS = 2:1; Prosolv^®^ SMCC 90) showed a very rapid release of SA during the acidic phase, leading to an artificial increase in the corresponding two dependences in Figure 3E,F by a factor of 5–6. The generally higher Δ*H*_CRG_/Δ*H*_SA_ ratios observed for the formulations with no binder (see Figure 3E) can be reasonably explained by the increased amount of CRG in these formulations, as suggested above. However, the similar principle (lower release of SA leading to a higher amount of SA ≈ Δ*H*_SA_ contained in the dissolving tablets) cannot in itself account for the large difference between the formulations containing calcium sulfate and the formulations with either SMCC or no binder. This unambiguously indicated that the content of available SA was further influenced by solid-state interaction with at least one tablet component. This was indeed true for the chitosan: the binding CHTS–SA interaction was reported in [30]. In addition, the study also reported on the inorganic anions disrupting the stability of the CHTS–SA complex. By adding these two intrinsic factors (formation of the CHTS–SA complex and its disruption by the calcium sulfate) to the systematic variations in the SA release rate during the acidic part of the dissolution process (detailed report in Section 3.2), the data in Figure 3E,F can be quantitatively explained.

### 2.2. Overall Swelling and Erosion Behavior of the Matrix Tablets and Their Morphological Characterization by SEM

In general, the matrix tablets faced various processes during the dissolution test. The tablets could absorb water or dissolve in the medium (which could result in various physico-chemical interactions between the tablet components and the medium, or between the particular tablet components), and the material of the tablet could be also eroded. Whereas in Section 3.1 we specifically addressed the initial stage of the dissolution process in the case of the water-insoluble model drug (SA), in the present section, the swelling and erosion of the tablets will be reported from an overall point of view. The corresponding quantifications of these processes were based on the “swelling index” and “remaining mass” parameters. The *swelling index* (SI) parameter described the ability of the matrix tablet to absorb the water or dissolution medium. It is an important property of controlled-release matrix tablets containing hydrophilic polymers. The tablet swelling and model drug release in the dissolution medium was especially influenced by the type and amount of the retarding polymer/mixture of polymers, as well as by the solubility of the used model drug. The release of a model drug with a poor solubility from the matrix tablets was usually significantly dependent on the matrix erosion. The parameter *remaining mass* (RM) characterized the loss of the tablet material during dissolution [31].

Due to the hydrophilic nature of the studied PEC-based matrix tablets, the swelling and erosion were the key processes that influenced the controlled model drug release. The swelling and erosion behaviors of all formulations were studied using a weighing method and described using SI and RM. As will be reported in Section 2.3, the model drug release (dissolution behavior of the studied tablets) was significantly influenced by the CRG/CHTS ratio, the type of the dry binder (microcrystalline cellulose or calcium sulfate), and the model drug solubility. The swelling and erosion of the studied tablets and pure binary mixture of CRG and CHTS upon exposure to the **acidic medium for 2 h** is shown in Figure 4A and Figure 5A. As can be seen, significant matrix swelling and slight matrix erosion were observed for all formulations. The increasing amount of CRG in the matrix more significantly influenced the matrix swelling compared to the erosion. The matrix swelling increased with the amount of CRG in the matrix (with the exception of the tablets without the dry binder and containing TH), as also determined based on the microscopic analysis (see Section 3.1). In the acidic medium, the sulfate groups of CRG were negatively charged and the free amino groups of CHTS were completely protonated (CHTS underwent degradation in the acidic medium). The electrostatic repulsions and effect of the solvation of the ionic group contributed to the swelling process.

As shown in Figure 4A, the lowest (≈345%, F5) and highest (≈750%, F17) swelling ratio in the acidic medium was observed for the tablets without dry binder, which contained SA as a model of a slightly water-soluble drug. Unlike SA, the presence of TH as a model of a highly water-soluble drug did not affect the swelling of the formulations without dry binder (F6, F12, and F18). Tablets F6, F12, and F18 swelled almost independently from the CRG/CHTS ratio, at a swelling rate of approx. 660%/2 h.

Because the type of dry binder (hydrophilic or hydrophobic) could significantly influence the swelling and the gel formation of the CRG/CHTS matrix tablets, two different types of the dry binders were used—silicified microcrystalline cellulose (SMCC) and CaSO_4_. SMCC (Prosolv^®^ SMCC90) is a combination of microcrystalline cellulose (MCC) and colloidal silicon dioxide (5%). SMCC is strongly hydrophilic excipient, and the hydroxyl groups of MCC allow formation of the hydrogen bonds. Therefore, the presence of SMCC in the formulations based on a physical mixture of CRG and CHTS can influence the process of swelling, the gel properties, and consequently the model drug release rate. Calcium sulfate is a hydrophobic dry binder. However, the presence of Ca^2+^ ions improves the swelling of ι-carrageenan due to forming the bivalent calcium bridges between the swollen particles [7]. In this study, no significant effect of the dry binder (nor the model drug solubility) on the degree of swelling was observed for the tablets with CRG/CHTS = 3/1. The degree of swelling (approximately 660%) was comparable to that of the tablets without the dry binder (Figure 4A). This meant that the swelling was primarily controlled by the CRG excess. For the tablets with CRG/CHTS = 2/1 (F7–F12), the effect of the Prosolv^®^ SMCC 90 presence was negligible (comparable SI with the dry-binder-free tablets), but a lower degree of swelling was observed for tablets containing CaSO_4_ (F9, F10). Therefore, it is reasonable to assume that for the tablets with CRG/CHTS = 2/1, the swelling was primarily influenced by the type of the dry binder, as the effect of the CRG excess decreased. For the tablets with CRG/CHTS = 1/1 (F1–F6), the degree of swelling of tablets with Prosolv^®^ SMCC 90 was comparable to that of the binder-free tablets (for both model drugs). Once CaSO_4_ was used, a significantly lower degree of swelling was observed (for both model drugs, approximately 500%). This was in good correspondence with the results reported in Section 3.1.

The effects of the CRG/CHTS ratio, type of dry binder used, and the model drug solubility on the swelling and erosion behavior of all studied matrix tablets in the **phosphate buffer at pH 6.8** are presented in Figure 4B and Figure 5B. The most significant changes compared to the acidic medium were observed for the tablets without the dry binder (F5, F6, F11, F12, F17, and F18) and with CaSO_4_ as the dry binder (F3, F4, F9, F10, F15, and F16). For the binder-free tablets, a significantly higher degree of swelling (≈827% in pH 6.8, ≈588% in pH 1.2) was observed for the CRG/CHTS ratio 2:1 in combination with SA (F11). When CaSO_4_ was used as the dry binder, the swelling of the tablets with the CRG/CHTS ratio 3:1 significantly decreased (in comparison with the acidic medium) almost independently from the model drug solubility (Figure 4B). This change in the swelling might have been influenced by a combination of the following factors: (1) highly deacetylated CHTS (85%) was soluble only in acidic solutions (up to pH of 6.5); (2) in a medium at pH 6.8, the ionic attraction between CHTS and CRG became weaker due to the lower amount of amino groups of CHTS being protonated (the p*K*_a_ of CHTS is around 6.5); and (3) due to the presence of CaSO_4_ in the tablets, calcium ions (in low concentration) were released in the medium, forming calcium salt of CRG—calcium salts of CRG do not dissolve in solutions at pH 6.8, but they exhibit the ability to swell (note that the formation of the CRG calcium salts was assumed based on the interpretation of the swelling behavior, and was not confirmed by the direct identification of these compounds). The rate of swelling of all studied formulations in the phosphate buffer at pH 6.8 significantly decreased (Appendix A). The effects of the CRG/CHTS ratio, the type of the dry binder, and the model drug solubility on the erosion behavior of all studied matrix tablets were insignificant (for both values of pH), as can be seen in Figure 5A,B. RM values higher than 80% were confirmed at pH 1.2, while slightly lower RM values were found at pH 6.8.

The above-mentioned results of the swelling and erosion tests confirmed that the studied CRG/CHTS-based matrix tablets have a promising potential for controlled model drug release, as the rapidity and intensity of the gel formation could be tailored by the CRG/CHTS ratio and by the used binder.

In addition to the swelling and erosion characteristics, the tablets that underwent the dissolution process were also characterized using the SEM technique. The SEM images of cross-section cuts of the F1, F11, F13, and F17 matrix tablets before dissolution, after dissolution in pH = 1.2, and after dissolution in pH = 6.8 are presented in Figure 6. The SEM images of the other formulations are shown in Appendix A).

The tablets before dissolution showed homogeneous dispersion of all their components throughout the tablet volume. The size of the particles (or their clusters) was usually below 0.1 mm; only the CaSO_4_ particles were larger, with a size regularly exceeding 0.2 mm. The tablets had no macroscopic pores (i.e., the maximum pore dimensions were in the order of 0.01 mm), and the overall visually perceived volume of the pores was very low compared to its increase in the tablets after dissolution. The dried tablets after dissolution had a relatively firm and compact crust on their surface, with a significantly increased overall hardness in comparison with the tablets before the dissolution. The core of the tablets after dissolution was porous and depleted of the soluble components (clusters of SA and TH). After the dissolution, the volume of the pores markedly increased, not only due to the depletion of the soluble components, but mainly due to the swelling and consequent drying of the excipients (the excipient particles shrank to their original size in the dried state, but the overall size of the swelled tablets remained almost unchanged during the drying, leading to the formation of large internal voids and cavities). After the dissolution and drying, the excipients not only retained their original size, but also their morphology, exhibiting no significant aggregation, agglomeration, or disintegration.

The SEM micrographs showed TH in a light hue (due to its heavier Cl atoms); however, this is not visible in the micrographs of tablets after dissolution (Figure 6). Considering that TH was only partially released from the tablets during the dissolution (as will be shown below in Section 2.3), the lack of its remains/residue in the corresponding SEM micrographs could indicate that all HCl was released from the tablets and that TH remained in its basic form or was bound to CRG (note that SA did not bind to CRG, as both compounds were negatively charged). It is also noteworthy that the structural morphologies of the tablets after dissolution (as viewed using SEM on the tablets’ cross-sections) were very similar for all formulations and pH values. The changes in the swelling indices, remaining masses, and other dissolution characteristics were not significantly reflected in the SEM micrographs.

### 2.3. In Vitro Model Drug Release

Model drug release from the studied CRG/CHTS-based matrix tablets was carried out to confirm the influence of in situ PEC formation on the model drug transport. The effects of the CRG/CHTS ratio (1:1; 2:1; 3:1), the type of dry binder (Prosolv^®^ SMCC 90, CaSO_4_) and the model drug solubility (TH, SA) on the dissolution behavior of the studied matrix tablets is summarized in Figure 7. The estimated parameters obtained from the fitting of the dissolution profiles to the Weibull model are shown in Table 2, including the coefficient of determination (*R*^2^) and absolute sum of squares of residuals (*RSS*).

It is well known that the release of highly water-soluble model drugs from hydrophilic matrices is predominantly controlled by water penetration and diffusional mass transport [32]. The release of slightly water-soluble model drugs is controlled more by polymer erosion than by diffusion [33]. When a combination of CHTS and an anionic polymer was used as the matrix system for the release of a slightly water-soluble model drug, the rate of erosion decreased with time due to the formation of the in situ PEC film on the tablet surface, and the influence of the diffusion increased. Meanwhile, the highly water-soluble model drug formed a large model drug concentration gradient in the swollen matrix, leading to the specific diffusion-controlled release mechanism. The key factors for the model drug release rate were the matrix swelling and the strength of the formed PEC film (the thickness of the diffusion layer) [11].

Our experiments confirmed the different release behaviors depending primarily on the model drug solubility (Figure 7). For the formulations containing TH as the model drug, a significantly lower dissolution was observed compared to the formulations with SA. As can be seen in Figure 7, the released amount of TH did not exceed 50% (except for the formulation F3), and this value was almost independent of the type of dry binder and the CRG/CHTS ratio. This fact was surprising and unexpected due to the excellent TH solubility and the high degree of swelling of these formulations in the acidic medium (Figure 4A). On the other hand, these results indicated the potential of the CRG/CHTS matrices for the controlled release of highly water-soluble model drugs. The problem of the low released amount of the model drug could be partially countered by a decrease in the compression force during the tablets’ preparation. As mentioned above, the water-soluble model drugs were mainly released via the diffusion through the gel layer. Therefore, the decrease in the TH release rate could indicate the low porosity of the gel layer for the TH molecules. The permeability of the dissolved model drug molecules/ions across the swollen polymeric network or film played a key role in the controlled model drug release. Due to the different sizes of TH (molecular weight: 299.84 g/mol) and SA (molecular weight: 138,12 g/mol) molecules, the difference in mass transport between these two model drugs was expected. Thus, the slower release of TH compared to that of SA, despite its high solubility, could be explained by its higher molecular weight [34]. Similar results were reported also by Li et al. [35]. In addition, a potential formation of the CRG–TH complex (the existence of which was reported by Aguzzi 2002 [36], which would cause retardation of the model drug release kinetics, could also have played a certain role. The potential formation of the CRG–TH complex might be responsible for the lowering of the TH melting onset (as displayed in Figure 1C), but it needs to be stated that no conclusive evidence for such interaction could be derived from the present results (leaving the lower TH permeability through the polymeric network as the most probable reason).

The highest release rate was confirmed for F5 (SA, without dry binder, CRG/CHTS = 1/1) and F7 (SA, Prosolv^®^ SMCC 90 as the dry binder, CRG/CHTS = 2/1) formulations. During the dissolution process in the acidic medium (2 h), almost 100% of SA was released. Therefore, these formulations were not suitable for the transport of the slightly water-soluble model drug into the intestine. On the other hand, the matrices containing calcium sulfate as the dry binder released a low amount of SA (20%, resp. 40%), probably due to the low solubility of the dry binder, resulting in a compact gel layer.

Based on the results obtained from the dissolution tests, only formulations F1, F11, F13, and F17 were chosen as potentially suitable for the transport of a slightly water-soluble model drug into the intestine because these matrices released, after 2 h in the acidic medium, maximally 25% of the model drug, and after 16 h in the medium at pH 6.8, almost all amount of the incorporated model drug was released.

### 2.4. Testing of Rheological Properties and Mucoadhesion

Based on the results reported in Section 2.1, Section 2.2 and Section 2.3, especially the dissolution profiles in Figure 7, the matrices represented by formulations F1, F11, F13, and F17 were chosen as suitable candidates for the transport of a slightly water-soluble model drug into the intestine. The formulations were tested for rheological behavior and mucoadhesive properties, which significantly determined the contact of the matrix tablet with the intestinal mucosa, the release of the model drug from the matrix, and thus the resulting local therapeutic effect of the preparation in non-specific inflammations of the intestinal mucosa.

The cohesiveness of the matrix tablets after exposure to pH 6.8 was determined using a shear rate test on an absolute rheometer. The obtained flow curves were fitted with the power law model. After one hour at pH 6.8, the lowest consistency of the gel layer was revealed in the F1 formulation (CRG/CHTS ratio 1:1, SMCC as dry binder). When the CRG/CHTS ratio was increased to 3:1 (F13), a significant increase in the consistency (almost five-fold) was observed. Matrix tablets F11 and F17 (without dry binder) containing 52.7% resp. 59.25% CRG had a significantly higher consistency compared to those with 25% CRG (F1), but comparable to a formulation containing 37.5% CRG (F13). We concluded that if the matrix tablets contained the hydrophilic filler SMCC, lower concentrations of CRG were sufficient to form a highly viscous gel layer. After 3 and 6 h at pH 6.8, only the initially low consistency of the F1 formulation was increased. For the other three tested matrix tablets, the consistency was not increased, or was even decreased, again depending on the presence of SMCC and on the CRG/CHTS ratio—see Figure 8.

Based on the performed rotational rheological tests, the following conclusions were derived. The consistency of the gel matrices and their changes during the exposure to pH 6.8 were influenced by the presence, resp. absence, of SMCC and CRG. For a matrix containing SMCC and only 25% CRG, the consistency increased from only about 200 Pa. s^*n*^ to about 1000 Pa. s^*n*^. The matrix with Prosolv^®^ SMCC 90 and a higher concentration of CRG (37.5%) has a consistency of 1077 Pa∙s^*n*^ after one hour of exposure, which then decreased to about half (611 Pa. s^*n*^) over time. For the matrices without Prosolv^®^ SMCC 90, but with twice the CRG content, the consistency of the gel layer was high after only 1 h in pH 6.8 buffer, and then decreased with time.

Figure 9 shows the values of the index of flow behavior *n* (−) that expressed the degree of non-Newtonian behavior; i.e., the sensitivity of the system to stress. If Newtonian systems with constant viscosity independent of shear rate have an *n* equal to 1, then the tested formulations with *n* values ranging from 0.0380 to 0.2618 are highly shear-sensitive non-Newtonian systems. However, the rheological behavior of the systems, the structure of which is significantly influenced by applied force/stress, are more appropriately tested using the oscillation method.

The viscoelastic properties of the matrix tablets after exposure to first an acidic medium for 2 h and then pH 6.8 for 6 h were determined using oscillation testing. As shown in Table 3, the values of the elastic modulus *G*′ were higher than the values of the viscous modulus *G*″ for all tested formulations, demonstrating that the bicoherent structure typical of solid-like gel systems was formed at pH 6.8. This was also confirmed by the phase angle values in the narrow range of 5.77° to 8.30°. Such values were typical for highly elastic systems. The results showed that the matrix tablets based on CRG and CHTS containing the water-insoluble model drug retained the nature of a viscoelastic solid even after being exposed to pH 1.2 for 2 h and pH 6.8 for 6 h. The bicoherent gel structure was not destroyed either by the gradual swelling of the polymer excipients or by reducing their concentration in the formulation. The influence of a hydrophilic dry binder was also not found.

Figure 10 compares the stiffness of the gel matrices, expressed by the complex modulus, which ranges from approximately 12,000 to 23,000 Pa. After 1 h in pH 6.8, the gel matrix containing Prosolv^®^ SMCC 90 and the CRG/CHTS ratio 1:1 showed the lowest stiffness. Formulations with a higher CRG/CHTS ratio and free of Prosolv^®^ SMCC 90 provided significantly stiffer gels. After 3 h in pH 6.8, all matrices exhibited high stiffness. After a total of 6 h in pH 6.8, the stiffness of the F1 gel matrix increased, while the stiffness of the other tested formulations decreased. In summary, the stiffness of the gel matrix containing SMCC and the CRG/CHTS ratio of 1:1 is the lowest after 1 h of exposure to pH 6.8 but increases with the exposure time of the pH 6.8 buffer. Formulations containing SMCC and the CRG/CHTS ratio of 3:1, and formulations without Prosolv^®^ SMCC 90 and the CRG/CHTS ratio of 3:1 and 2:1 form solid gel matrices after only one hour and a significant decrease in stiffness does not occur until after 6 h.

Mucoadhesion of the matrix tablets after exposure to pH 6.8 for 1, 3, and 6 h was expressed as the maximum forces required for the detachment of the sample from the substrate, relative to the contact area *F*/*A* (mN.mm^−2^)—see Figure 11. After one hour at pH 6.8, the mucoadhesion was higher in the gel matrices containing Prosolv^®^ SMCC 90 (F1 and F13) than in the matrices without Prosolv^®^ SMCC 90 (F11 and F17) and in matrices with a lower CRG/CHTS ratio. This phenomenon was not unexpected and can be explained by the spreading theory of bioadhesion: a material with lower viscosity or lower structure stiffness adheres better to the substrate, as it is able to spread better and make intimate contact with the substrate. If the consistency decreased with the exposure time to pH 6.8 and the structure of the solid gel matrices relaxed, the adhesion to the model mucin increased, as was seen, for example, with formulations F13 and F17 after 6 h at pH 6.8.

The in vitro detachment force methods known as the tensile methods are the most widely employed techniques for investigation of the adhesive interactions between a model mucosal membrane and a formulation. Although they are most often evaluated in terms of the maximum force required for removal of the adhesive sample from the substrate [37], an interesting parameter is definitely the time required for the force to be reduced by 90% of its peak value (reflecting the adhesion time). As seen in Figure 12, the mucoadhesion was longer in the gel matrices with a higher consistency and stiffness, and also increased with the residence time of the matrix in the pH 6.8 environment.

Gel matrices with higher coefficient of consistency and stiffness of the gel structure showed a lower adhesive strength to model mucin in the in vitro tensile tests caused by more difficult spreading on the model substrate. As the gel structure relaxed, the adhesive strength increased. Likewise, the time of mucoadhesion increased with the relaxation of the structure and the exposure time to pH 6.8.

It should be noted that for the present matrix tablets, the rheological properties and mucoadhesive behavior were determined in vitro under experimental conditions practically identical to those experienced in vivo. It can be assumed that the mucoadhesiveness of the matrix tablets detected by this in vitro test will also manifest itself in vivo. Such correlation between the in vitro performed mucoadhesive tensile tests and real in vivo behavior of the matric tablets was confirmed, e.g., in [38,39].

## 3. Materials and Methods

### 3.1. Materials

ι-carrageenan (Merck KG&A, Darmstadt, Germany) and chitosan of a medium molecular weight and an 85% degree of deacetylation (JBICHEM, Shanghai, China) were used as the agents to form the polyelectrolyte-based matrix system. Silicified microcrystalline cellulose (Prosolv^®^ SMCC 90; JRS PHARMA GmbH & Co. KG, Rosenberg, Germany) or calcium sulfate dihydrate (Compactrol^®^; JRS PHARMA GmbH & Co. KG, Germany) were used as a dry binder, and sodium stearyl fumarate (Pruv^®^; JRS PHARMA GmbH & Co. KG, Germany) was used as a lubricant. Tramadol hydrochloride (Sigma Aldrich Chemie GmbH, Taufkirchen, Germany) was chosen as a model highly water-soluble drug, and salicylic acid (NOVACYL SAS, Écully, France) as a model slightly water-soluble drug. Mucin from porcine stomach Type II was purchased from Merck (Merck KG&A, Darmstadt, Germany) and used as a model substrate for adhesion testing.

Redistilled water and chemicals of an analytical grade (HCl, NaCl, NaOH, and KH_2_PO_4_; Lach-Ner s.r.o., Neratovice, Czech Republic) were used for the preparations of the dissolution media and the standard solution of the model drugs.

### 3.2. Methods

#### 3.2.1. Preparation of the Tableting Materials

The study employed 18 tableting materials, the compositions of which are shown in Table 1. A mixing cube (KB 15S; Erweka GmbH, Germany) was used for the mixture preparation. Tableting materials were prepared using graded mixing. First, mixtures of CRG and CHTS at ratios of 1:1, 1:2, and 1:3 were prepared by mixing of substances for a period of 2.5 min. These mixtures were mixed with silicificated microcrystalline cellulose (SMCC) or calcium sulfate dihydrate for 2.5 min. Drugs were added for other 2.5 min of mixing. Finally, natrium stearyl fumarate was added for a period of mixing of 2 min.

#### 3.2.2. Preparation of the Tablets

Tablets were compressed using a T1 FRO 50 TH.A1K Zwick/Roell device (Zwick GmbH & Co. KG, Ulm, Germany) by means of a special die with a lower and an upper punch. The rate of compression was 40 mm/min, the preload was 2 N, and the rate of preload was 2 mm/s. The tablets were cylindrical without facets and had a diameter of 7 mm and a weight of 0.100 ± 0.001 g.

The compression force was different for each tableting material, but such that the tablet tensile strength was in the range of 1–1.2 MPa. For each tableting material, 100 tablets were compressed and then used for testing.

#### 3.2.3. Swelling and Erosion of the Tablets

Testing of the swelling and erosion behavior of the studied matrix tablets was carried out using a dissolution apparatus (Sotax AT 7 Smart, Allschwill, Switzerland) using the rotating-basket method. The swelling index (SI) and remaining mass (RM) were determined using a gravimetric analysis at 37 ± 0.5 °C. All studied tablets were weighed before the dissolution test (dry tablets). Then, the tablets were submerged in 900 mL of the acidic dissolution medium (pH 1.2) for 2 h. After 2 h, the baskets with tablets were transferred into 900 mL of phosphate buffer (pH 6.8). The dissolution test continued for 10 h in phosphate buffer. The tablets were withdrawn from the dissolution medium, the excess liquid was removed, and tablets were weighed at the following significant time intervals: (1) after 2 h in acidic medium; (2) after 3 h in phosphate buffer; and (3) after completing the dissolution test. For the RM determination, the tablets also were weighed after drying for 24 h. The drying was executed in a pre-heated oven at 40 °C.

Based on the values obtained via the gravimetric analysis, the SI and RM were determined according to the equations mentioned below. The calculated indices of the individual tablets were compared.

##### Swelling Index

The swelling index (SI) was expressed as the ratio of a weight of absorbed water and a weight of the dry tablet before dissolution testing. The SI was calculated according to the following equation:(1)$$\mathrm{Swelling}\;\mathrm{index}\;\left[\%\right]=\frac{W_{t}-W_{0}}{W_{0}}\times 100$$
where *W*_0_ (g) is the weight of a dry tablet before dissolution testing and *W_t_* (g) is the weight of a swollen tablet at time *t* [11,40,41,42]. The swelling study was carried out in triplicate for all studied tablets.

##### Remaining Mass

The calculation of remaining mass is described by the following equation [11,43]:(2)$$\mathrm{Remaining}\;\mathrm{mass}\;\left[\%\right]=\left(\frac{W_{d\;}}{W_{0}}\right)\times 100$$
where *W*_0_ (g) is the weight of a dry tablet before dissolution testing and *W_d_* (g) is the weight of a dried tablet after dissolution testing. The study was carried out in triplicate for all studied tablets.

##### Visual Quantification of the Swelling and Medium Absorption

For the selected formulations, optical microscopy was used to further quantify the swelling kinetics. Using a Dino-Lite Edge 3.0 microscope, the tablets taken out of the dissolution medium (pH 1.2, temperature 37 °C, stirring at 100 rpm) at pre-selected times (1, 5, 30, 60 and 120 min) were photographed from the top view and on the vertical cross-section. Using the dimension calibration standard, the diameter and thickness of the swollen tablets, as well as the thickness of the dry tablet cores, were evaluated.

#### 3.2.4. Differential Scanning Calorimetry (DSC)

In order to detect the presence of the bond-free CRG in the tablets, differential scanning calorimetry was used as an identification technique for the major components of the studied formulations (CRG—based on the exothermic decomposition peak at 190–205 °C; SA—based on the melting peak at ~175 °C; and TH—based on the melting peak at ~190 °C). Note that the above-mentioned temperature ranges were determined using DSC measurements of the individual pure components, and later confirmed by observing similar effects for the matrix tablets containing these components. The DSC measurements were performed by using a heat-flow differential scanning calorimeter Q2000 (TA Instruments, New Castle, DE, USA) equipped with an autosampler, an RCS90 cooling accessory, and T-zero technology. Calibration of the instrument was performed using In, Zn, and H_2_O. Dry N_2_ at flow rate of 50 cm^3^·min^−1^ was used as a purge gas. The samples were taken directly from the dissolution-processed tablets taken out of the corresponding medium at pre-selected times. For each tablet, the samples were removed from the tablet edge, the center of the top tablet side, and the center of the whole tablet. The samples with masses of approx. 20 mg were inserted into low-mass T-zero pans (no lid was used in order for the hydration/decomposition products to by freely removed by the purge gas). The DSC measurements were designed as simple heating scans performed at heating rate of *q^+^* = 30 °C·min^−1^ from 25 to 300 °C. Integration of the DSC signals/peaks was conducted using a (physically meaningful) tangential area-proportional baseline.

#### 3.2.5. Scanning Electron Microscopy (SEM)

The VEGA3 SBU compact scanning electron microscope (Tescan, Brno, Czech Republic) was used to evaluate the homogeneity and cross-section profile of tablets before and after the dissolution. An acceleration voltage of 15 kV, a backscattered electron (BSE) detector, and low vacuum mode (10 Pa, N2) were applied. Note that the used BSE detector had the advantage of an increased sensitivity to the atomic mass number (the higher the number, the lighter the color hue). The cuts were performed using a scalpel to cut through the middle of the tablets. After fixing the tablets to circular aluminum discs using carbon-based, electrically conductive, double-sided adhesive discs (Agar Scientific Ltd, Stansted, UK), the tablets’ cross-section areas were blown over using compressed air, and the aluminum discs were placed into the microscope chamber. The SEM investigation was mainly focused on the center of the tablets’ cross-sections.

#### 3.2.6. In Vitro Drug-Release Studies

Drug-release tests were carried out with a dissolution apparatus (Sotax AT 7 Smart, Allschwill, Switzerland) using the rotating-basket method. All dissolution tests were performed according to the *European Pharmacopoeia* 10th Edition (*Ph. Eur.*, *European Pharmacopoeia*, 2020). Two different dissolution media prepared according to the *Ph. Eur.* were used: (1) an acidic medium at pH 1.2 (HCl with an adjustment of ionic strength using NaCl); (2) a phosphate buffer at pH 6.8. All tests were carried out for 18 h at a stirring rate of 100 rpm. The temperature was maintained at 37 ± 0.5 °C. Firstly, the tablets were put into the baskets and submerged in the dissolution vessels containing 900 mL of the acidic medium. Dissolution in the acidic medium was performed for 2 h. After 2 h in the acidic medium, the baskets with tablets were transferred into the dissolution vessels containing 900 mL of the phosphate buffer at pH 6.8. The dissolution testing continued for additional 16 h in the medium at pH 6.8. This method was used to simulate the transit of the tablets through the gastrointestinal tract. During the dissolution tests at predetermined time intervals, 3 mL aliquots of the dissolution medium were automatically withdrawn and filtered. Each experiment was performed with six tablets and one blank sample. In the blank samples, the amount of the model drug was replaced with the dry binder (calcium sulfate or Prosolv^®^ SMCC 90). For tablets containing only the physical mixture of CRG and CHTS without a dry binder (F5, F6, F11, F12, F17, and F18), the blank samples were prepared using the physical mixture of CRG and CHTS with a lubricant. The dissolution tests were carried out in triplicate for each formulation studied, and standard deviations were calculated. In all dissolution tests, the released amount of the model drug was determined using UV–Vis spectrometry, and the dissolution profiles obtained were evaluated using suitable kinetic models.

#### 3.2.7. Determination of the Drug-Release Amount Using UV–V Is Spectrometry

An HP Agilent 8453 spectrophotometer (Agilent Technologies, Santa Clara, CA, USA) was used to determine of the model drug (TH or SA) concentration in the dissolution samples [44,45,46,47] The absorbance values of the samples withdrawn at the predetermined times were measured against the corresponding blank sample using the fixed-wavelength method. A wavelength of 272 nm for TH and 298 nm for SA together with three-point background correction were applied. The used wavelengths corresponded to the absorption maximum of the model drugs. The validity of the Lambert–Beer law was verified in the expected range of the drug concentrations. To transform the absorbance values into the concentrations and percentages, the calibration-curve method was used.

#### 3.2.8. Non-Linear Regression Analysis of the Dissolution Profiles

In order to quantitatively evaluate the released amount of the model drug (TH or SA) from the studied CRG-CHTS formulations, the obtained dissolution profiles were fitted to the Weibull model expressed by the following equation:(3)$$M_{t\left(l\right)}=M_{\infty }\left(1-\exp\left(-k_{w}t^{\beta }\right)\right)$$

In Equation (3), $M_{t\left(l\right)}$ is the amount of the drug released in time *t*, $M_{\infty }$ is the maximum releasable amount of the drug in infinite time (it should be equal to the absolute amount of the drug incorporated into a matrix tablet at the time *t* = 0), and *k*_w_ is a constant of the Weibull model with unit *time^−β^*. Parameter *β* characterizes the shape of the exponential curve. GraphPad Prism 9.3.1 (GraphPad Software, San Diego, CA, USA) was employed for the non-linear regression analysis.

#### 3.2.9. Testing of Rheological Properties

The flow and viscoelastic behavior of the formulations F1, F11, F13, and F17 after exposure to the buffer pH of 1.2 (2 h), and subsequently to pH 6.8 (6 h), were tested using a Kinexus Pro+ (Malvern, UK) rotational rheometer with a Peltier plate cartridge and a PU 20 measuring system and the standard pre-configured sequences *Toolkit_V005 Shear Rate Ramp—Alternative Flow Curve* and *rSolution_0008 characterising gel properties using oscillatory testing* in the rSpace software version 1.76. A standard loading was used to ensure that the samples were subjected to a consistent and controllable loading protocol. The measurements were performed at 37.0 °C. A linear part of the flow curves obtained in a shear rate range between 0.1 and 100 s^−1^ was fitted with a power law model:(4)$$\tau =K\;D^{n}$$
where τ is shear stress (Pa), *D* is shear rate (s^−1^), *K* is the coefficient of consistency (Pa s^*n*^), and *n* is the index of flow behavior (unitless). The coefficient of consistency (or simply consistency) is numerically equal to the viscosity at a shear rate of 1 s^−1^. The index of flow behavior is a measure of non-Newtonian behavior (*n* = 1 is for a Newtonian system; *n* > 1 means shear thickening; *n* < 1 means shear thinning). All the measurements were done in triplicate with newly loaded samples, and the mean and standard deviation were calculated.

The oscillation test was performed with a gap of 1 mm and shear rate range between 0.01% and 100% at a frequency of 1 Hz. The elastic modulus *G*′, viscous modulus *G*″, complex modulus *G**, and phase angle *δ* were used for evaluation of the formulations’ viscoelastic properties. Here, *G*′ > *G*″ indicates a solid-like material, while *G*″ > *G*′ applies to a liquid-like material. Ideally, an elastic material has *δ* = 0°, an ideally viscous material has *δ* = 90°, and values between these limits characterize different degrees of viscoelasticity. Stiffness is reflected in the complex modulus *G**, with higher values indicating a stiffer structure, while the phase angle *δ* indicates the degree of elasticity and hence springiness of the structure. All the measurements were done in triplicate with newly loaded samples, and the mean and standard deviation were calculated.

#### 3.2.10. Testing of Mucoadhesion

Testing of mucoadhesion was performed on formulations F1, F11, F13, and F17. The tablets were immersed in the pH 1.2 buffer for two hours and then in the pH 6.8 buffer for 6 h. The assay was performed after 1 h, 3 h, and 6 h treatment at pH 6.8. Mucoadhesive properties were measured using a Kinexus rotational rheometer with a Peltier plate cartridge using matched PU 20 and the standard pre-configured sequence *rSolution_0019 Determination of pressure sensitive tack and adhesion using axial measurements test* in the rSpace for Kinexus software version 1.76. Mucin from porcine stomach Type II (Merck, Prague, Czech Republic) hydrated with optimal amount of the PBS at pH 6.8 was used as a model substrate; 150 mg of mucin substrate was spread on the area of the upper and lower rheometer geometry, and the tested sample was placed in the center of the lower geometry. During the tensile test, a contact time of 10 s, a consolidation force of 1 N, and an upper geometry detachment speed of 10 mm·s^−1^ were used. The peak in normal force *F*_max_ (N) necessary for the detachment of the model substrate from the sample was converted to the contact area delineated by the upper geometry. Thus, the mucoadhesion was expressed in (mN∙mm^−2^). All the measurements were taken five times with newly loaded samples, and the mean value and standard deviation were calculated.

## 4. Conclusions

Matrix tablets based on chitosan–carrageenan biopolymeric excipients were tested for their utilization in extended-release dosage forms and possible model drugs targeting the intestine. Salicylic acid and tramadol hydrochloride were used as the model drugs. The tablets were found to significantly swell (by ~500–800%) and form a cohesive gel throughout their volume. The swelling was found to be primarily driven by the carrageenan presence. The DSC analyses indicated the initiating formation of the chitosan–carrageenan complex during immersion in the acidic medium. The formation of this polymeric complex appeared to be almost instantaneous after the sample wetting, and the presence of H^+^ ions (acidic medium) appeared to significantly accelerate this formation. The gel properties were found to be determined not only by the chitosan–carrageenan ratio, but also, to a smaller extent, by the presence of microcrystalline cellulose and CaSO_4_. The chitosan–carrageenan gels were found to be permeable (via the standard diffusion mechanism) for the smaller model drug molecules (salicylic acid in the present case). On the other hand, in case of the tramadol hydrochloride (a significantly larger molecule), the released amount of the model drug was (despite rapid wetting and significant tablet erosion) only ~40%. Hence, the apparent molecular weight dependence of the chitosan–carrageenan gel permeability must be taken into an account (together with the potential chemical interactions, such as SA + CHTS or TH + CRG) for the practical applicability of these formulations. Formulations with different model drugs must be checked on an individual basis. Based on the dissolution tests, several formulations (F1, F11, F13, and F17) were chosen as suitable candidates for usage in extended-release matrix tablets, with the common denominator being either the high carrageenan content or the suitable CRG/CHTS/Prosolv^®^ SMCC 90 ratio. It should be noted that the formation of the chitosan–carrageenan polyelectrolyte complex may be an elegant and beneficial alternative solution for the targeting of the matrix tablet in the intestine. For this reason, the selected formulations (F1, F11, F13, and F17) were further subjected to extensive testing of their rheological behaviors and mucoadhesive properties. Gel matrices with a higher coefficient of consistency and stiffness of the gel structure showed a lower adhesive strength to model mucin in the in vitro tensile tests, as caused by more difficult spreading on the model substrate. As the gel structure relaxed, the adhesive strength increased. Likewise, the time of mucoadhesion increased with the relaxation of the structure and the exposure time to pH 6.8. The two most suitable profiles were obtained for the formulations F1 (which had a very high initial adhesivity that significantly decreased in the late stage of 720 h) and F11 (which had an above-average initial adhesivity that increased to high levels with prolonged exposure to the pH 6.8 conditions). Considering the almost triple adhesion times obtained for the formulation F11 (compared to F1), this matrix composition (CRG/CHTS = 2:1, no dry binder) is recommended for further testing and potential usage in intestine-targeted extended-release matrix tablets.

The chitosan–carrageenan complex prevented unintended model drug release in the stomach, and subsequently, the mucoadhesive gel matrix tablet provided sustained model drug release in the small intestine. Tablets with a chitosan–carrageenan polyelectrolyte complex are a promising matrix system for the treatment of non-specific intestinal inflammations.

## Figures and Tables

**Figure 1 pharmaceuticals-15-00980-f001:**
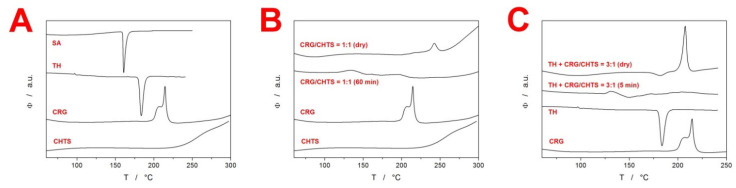
(**A**) DSC curves obtained at 30 °C·min^−1^ for the major pure components of the present matrix tablets. (**B**) DSC curves for the dry and wetted (within the standard acidic dissolution procedure) binary CRG/CHTS tablets and their pure components. (**C**) DSC curves for the dry and wetted (within the standard acidic dissolution procedure) example TH-containing formulation (F18) and their pure substances.

**Figure 2 pharmaceuticals-15-00980-f002:**
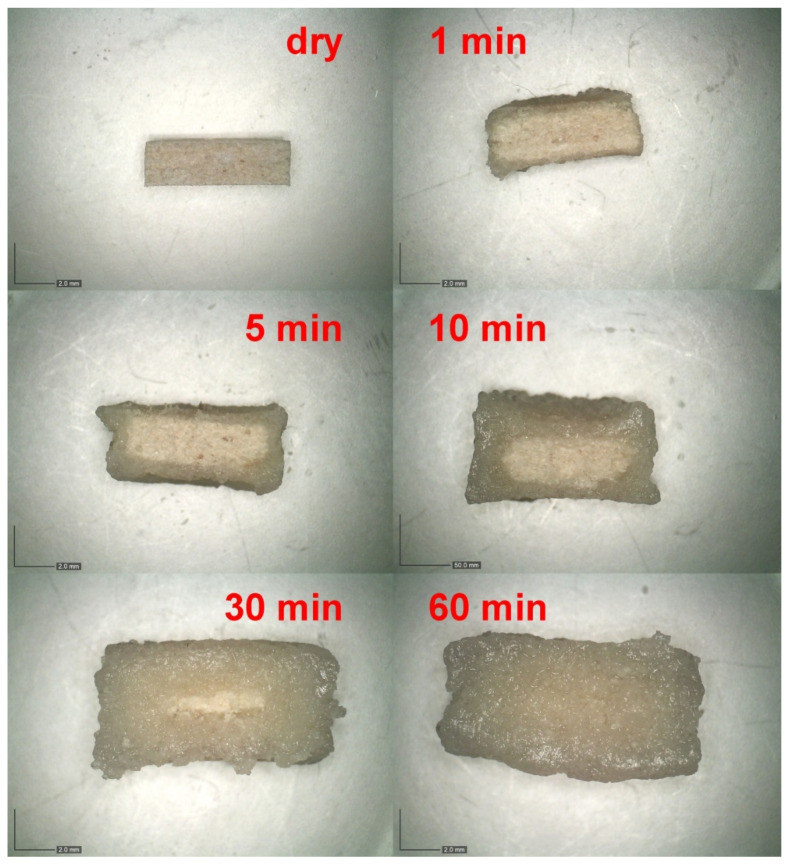
Example of the cross-section micrographs taken for the dissolution in acidic medium performed for the formulation F13 (CRG/CHTS = 3:1; Prosolv^®^ SMCC 90 used as the dry binder).

**Figure 3 pharmaceuticals-15-00980-f003:**
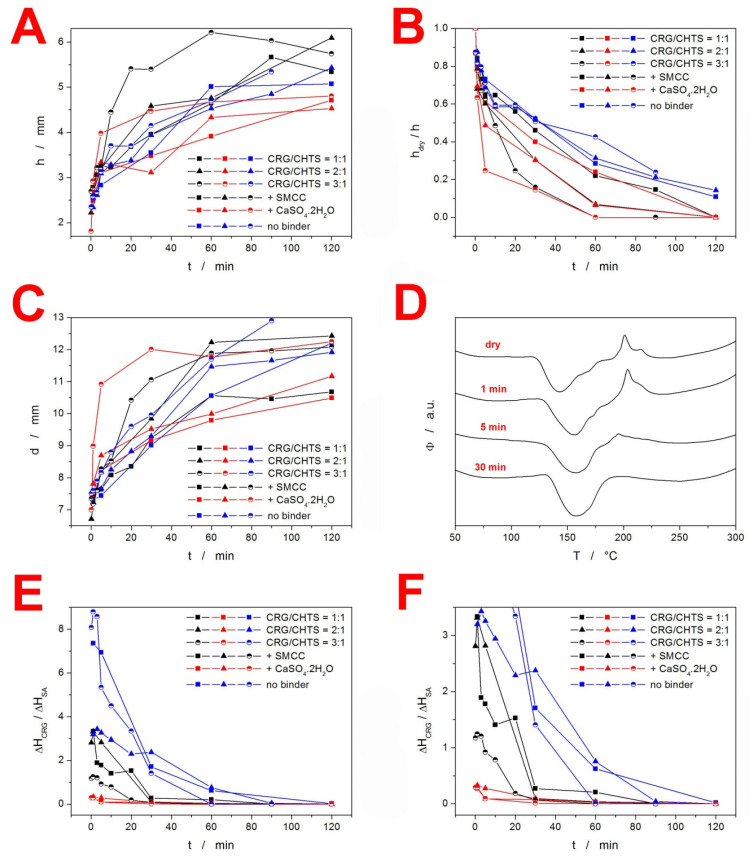
(**A**) Evolution of the tablet height *h* (measured in the cross-section view) during the dissolution in the acidic medium. (**B**) Evolution of the *h*_dry_/*h* ratio (where *h*_dry_ is the height/thickness of the dry tablet core) during the dissolution in the acidic medium. (**C**) Evolution of the tablet diameter *d* (measured in the top view) during the dissolution in the acidic medium. (**D**) Example DSC curves for the samples taken from the “dry” F15 tablet core. Duration of the dissolution in the acidic medium is indicated. (**E**) Ratio between enthalpies of the two characteristic DSC peaks (melting for SA and decomposition for CRG). (**F**) A zoomed-in view of the graph in (E). The error bars in all graphs are lower than four times the magnitude of the points.

**Figure 4 pharmaceuticals-15-00980-f004:**
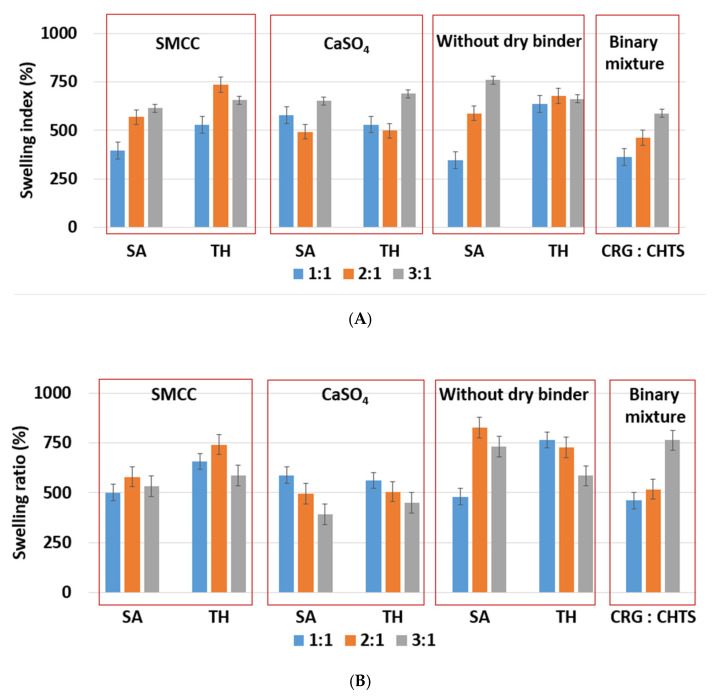
(**A**) The effect of CRG/CHTS ratio (1:1; 2:1; 3:1), type of dry binder (Prosolv^®^ SMCC 90, CaSO_4_), and model drug solubility (TH, SA) on swelling behavior of studied matrix tablets in acidic medium (2 h). (**B**) The effect of the CRG/CHTS ratio (1:1; 2:1; 3:1), type of dry binder (Prosolv^®^ SMCC 90, CaSO_4_), and model drug solubility (TH, SA) on the swelling behavior of the studied matrix tablets in phosphate buffer at pH 6.8 (2 h in acidic medium then 2 h in phosphate buffer).

**Figure 5 pharmaceuticals-15-00980-f005:**
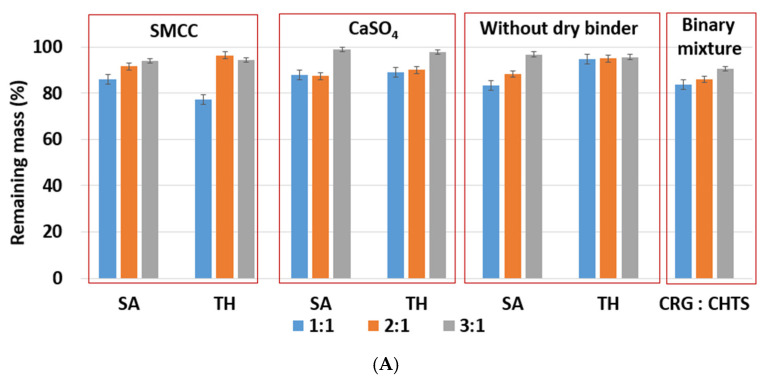
(**A**) The effect of ratio CRG/CHTS (1:1; 2:1; 3:1), type of dry binder (Prosolv^®^ SMCC 90, CaSO_4_), and model drug solubility (TH, SA) on the erosion behavior of the studied matrix tablets in acidic medium (2 h). (**B**) The effect of the CRG/CHTS ratio (1:1; 2:1; 3:1), type of dry binder (Prosolv^®^ SMCC 90, CaSO_4_) and model drug solubility (TH, SA) on the erosion behavior of the studied matrix tablets in phosphate buffer (2 h in acidic medium then 2h in phosphate buffer).

**Figure 6 pharmaceuticals-15-00980-f006:**
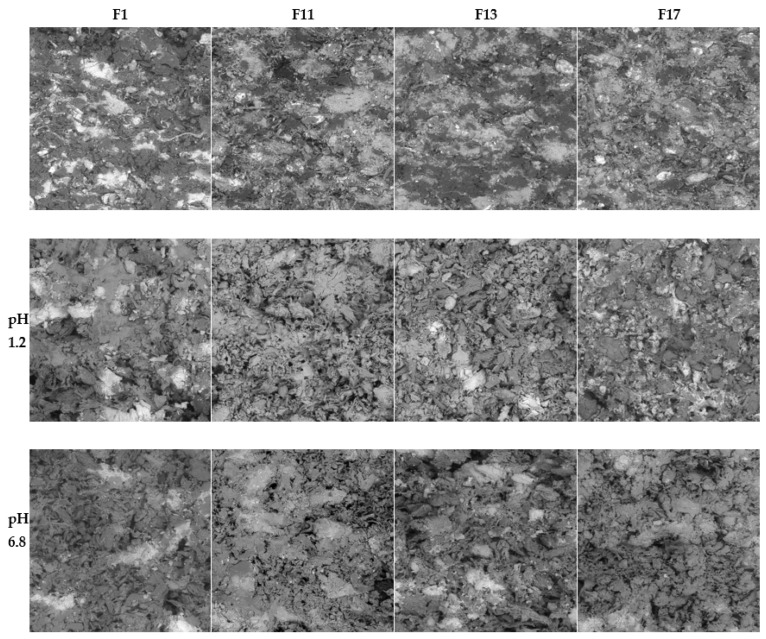
Cross-section cuts of the F1, F11, F13, and F17 matrix tablets before dissolution (**upper row**), after dissolution in pH = 1.2 (**middle row**), and after dissolution in pH = 6.8 (**bottom row**) as viewed using SEM. The dimensions of each micrograph are 1 mm × 1 mm.

**Figure 7 pharmaceuticals-15-00980-f007:**
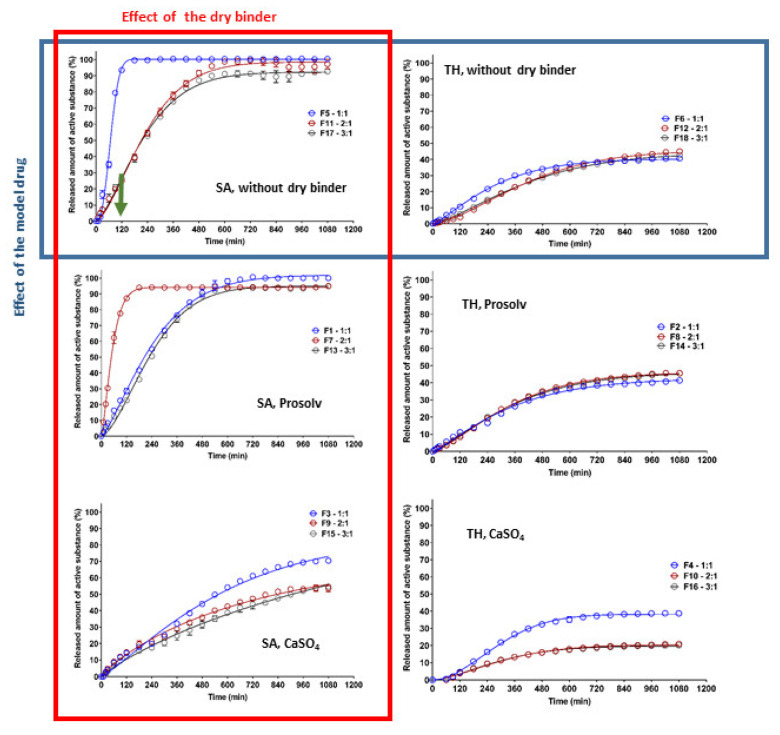
The effect of CRG/CHTS ratio (1:1; 2:1; 3:1), type of dry binder (Prosolv^®^ SMCC 90, CaSO_4_), and model drug solubility (TH, SA) on the dissolution behavior of studied matrix tablets.

**Figure 8 pharmaceuticals-15-00980-f008:**
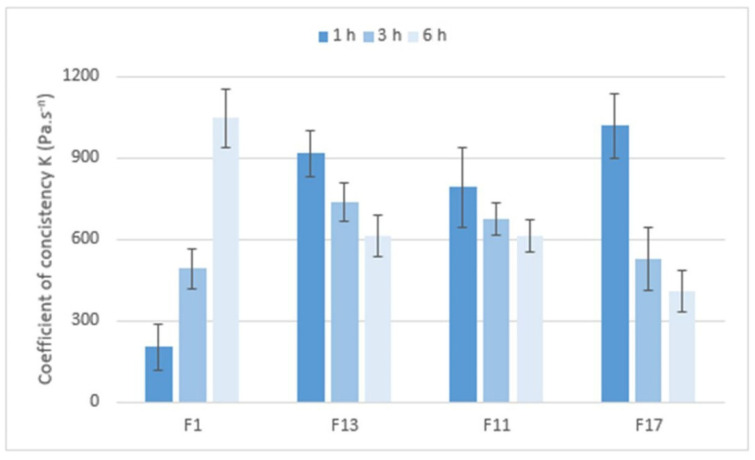
The values of the coefficient of consistency *K* (Pa. s^n^) for matrix tablets F1, F11, F13, and F17.

**Figure 9 pharmaceuticals-15-00980-f009:**
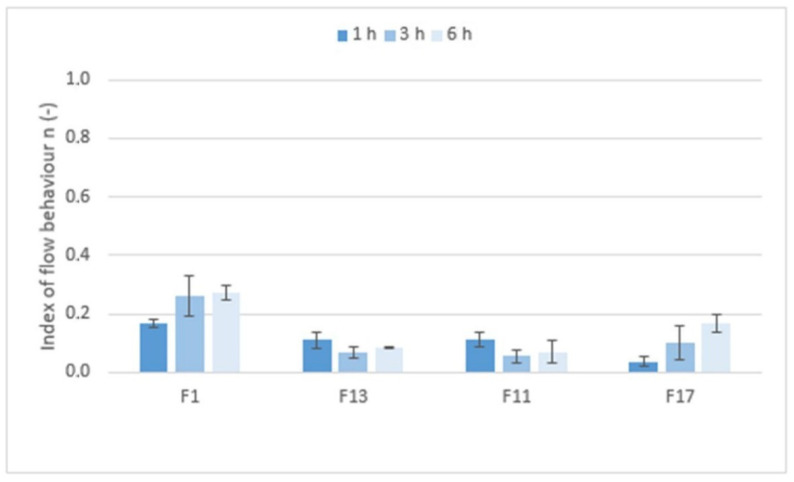
The values of the index of flow behavior *n* (−) for matrix tablets F1, F11, F13, and F17.

**Figure 10 pharmaceuticals-15-00980-f010:**
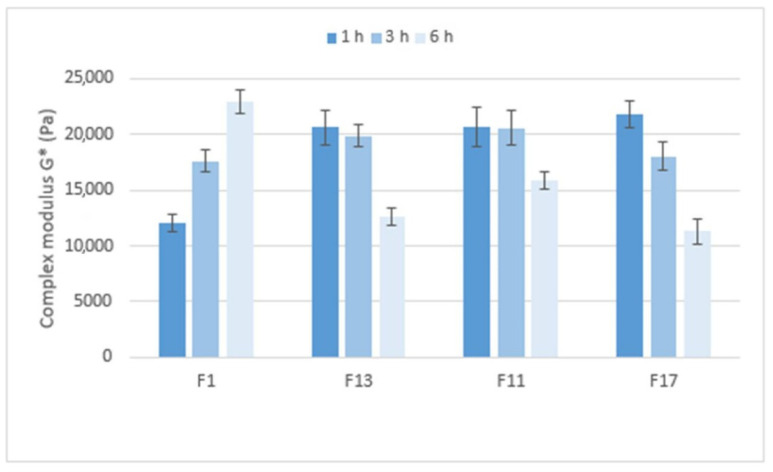
The values of the complex modulus *G** (Pa) for matrix tablets F1, F11, F13 and F17.

**Figure 11 pharmaceuticals-15-00980-f011:**
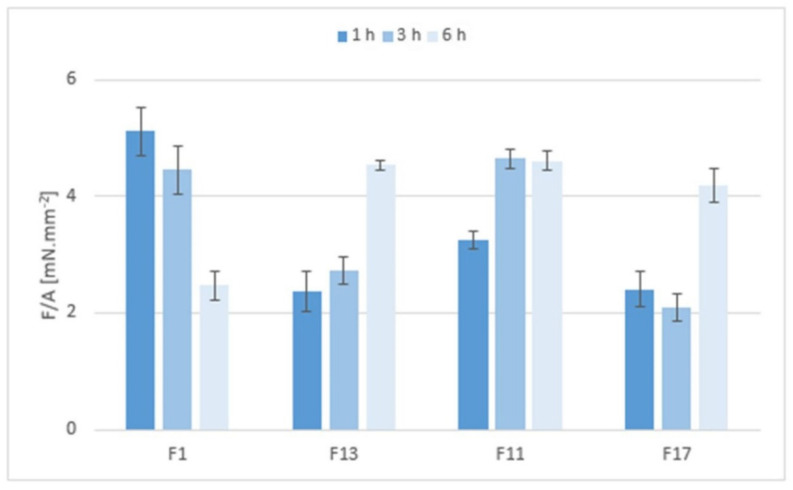
The mucoadhesivness of the matrix tablets F1, F11, F13, and F17 expressed as maximal detachment force *F* (mN) relative to the contact area *A* (mm^2^).

**Figure 12 pharmaceuticals-15-00980-f012:**
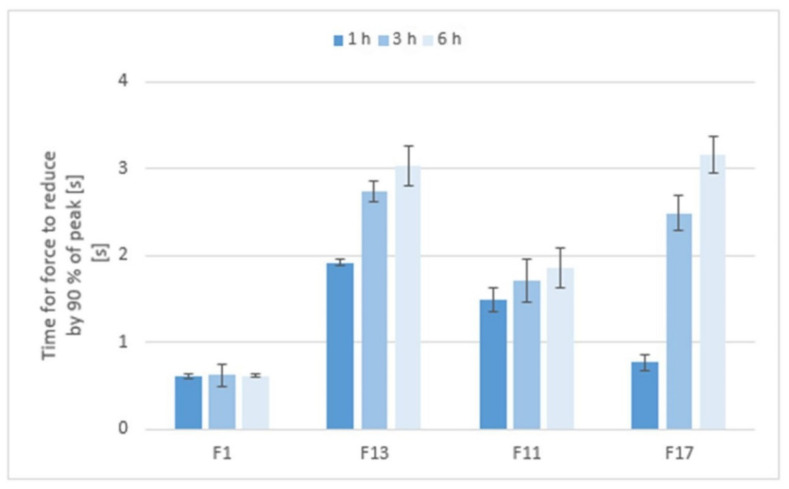
The adhesion time of matrix tablets to a model mucin substrate during an in vitro tensile test.

**Table 1 pharmaceuticals-15-00980-t001:** Compositions of studied formulations (wt %).

CRG/CHTS	Sample	CRG	CHTS	TH	SA	SMCC	CaSO_4_	Pruv^®^
1:1	F1	25	25	-	20	29	-	1
	F2	25	25	20	-	29	-	1
	F3	25	25	-	20	-	29	1
	F4	25	25	20	-	-	29	1
	F5	39.5	39.5	-	20	-	-	1
	F6	39.5	39.5	20	-	-	-	1
2:1	F7	33.3	16.7	-	20	29	-	1
	F8	33.3	16.7	20	-	29	-	1
	F9	33.3	16.7	-	20	-	29	1
	F10	33.3	16.7	20	-	-	29	1
	F11	52.7	26.3	-	20	-	-	1
	F12	52.7	26.3	20	-	-	-	1
3:1	F13	37.5	12.5	-	20	29	-	1
	F14	37.5	12.5	20	-	29	-	1
	F15	37.5	12.5	-	20	-	29	1
	F16	37.5	12.5	20	-	-	29	1
	F17	59.25	19.75	-	20	-	-	1
	F18	59.25	19.75	20	-	-	-	1

Explanation of abbreviations used: CRG—ι-carrageenan; CHTS—chitosan; TH—tramadol hydrochloride; SA—salicylic acid; SMCC—silicificated microcrystalline cellulose; CaSO_4_—calcium sulfate dihydrate; Pruv^®^—sodium stearyl fumarate.

**Table 2 pharmaceuticals-15-00980-t002:** Non-linear regression analysis of the dissolution profiles—estimated parameters obtained from fitting the dissolution profiles to the Weibull model.

Formulation	$\mathrm{Weibull}\;\mathrm{Model}\;M_{t\left(l\right)}=M_{\infty }\left(1-\exp\left(-k_{w}t^{\beta }\right)\right)$			
	$\begin{array}{c} (k_{w}\;\pm \;\mathrm{SD})\;\times \;10^{3}\\ \left(\mathrm{mi}n^{-\beta }\right) \end{array}$	*β* ± *SD*	*RSS*	*R* ^2^
F1	0.61 ± 0.11	1.32± 0.03	182.7	0.9973
F2	0.80 ± 0.19	1.21 ± 0.04	55.8	0.9947
F3	0.43 ± 0.28	1.21 ± 0.13	984.6	0.9684
F4	0.02 ± 0.01	1.86 ± 0.07	60.2	0.9951
F5	0.02 ± 0.01	2.45 ± 0.10	215.9	0.9966
F6	0.76 ± 0.09	1.26 ± 0.02	14.6	0.9986
F7	4.21 ± 0.31	1.35 ± 0.02	32.2	0.9992
F8	0.30 ± 0.03	1.37 ± 0.02	10.6	0.9992
F9	2.77 ± 0.48	0.89 ± 0.04	102.3	0.9938
F10	0.11 ± 0.06	1.54 ± 0.09	39.2	0.9882
F11	0.22 ± 0.06	1.51 ± 0.05	333.1	0.9951
F12	0.08 ± 0.01	1.53 ± 0.03	17.3	0.9987
F13	0.16 ± 0.04	1.56 ± 0.04	162.3	0.9973
F14	0.51 ± 0.08	1.27 ± 0.03	23.2	0.9982
F15	1.41 ± 0.19	0.90 ± 0.05	102.7	0.9936
F16	0.12 ± 0.07	1.55 ± 0.11	55.3	0.9826
F17	0.37 ± 0.09	1.43 ± 0.04	241.0	0.9959
F18	0.22 ± 0.03	1.38 ± 0.02	13.0	0.9989

**Table 3 pharmaceuticals-15-00980-t003:** The values of complex modulus *G**, elastic modulus *G*′, and viscous modulus *G*″ a phase angle δ for matrix tablets F1, F11, F13, and F17.

Formulation	Time (h)	*G** ± SD (Pa)	*G*′ ± SD (Pa)	*G*″ ± SD (Pa)	δ ± SD (°)
F1	1	12,080 ± 791	11,950 ± 965	1744 ± 97	8.30 ± 0.44
	3	17,610 ± 927	27,470 ± 1881	2813 ± 167	5.77 ± 0.27
	6	22,900 ± 1054	18,720 ± 1514	2551 ± 139	7.76 ± 0.51
F11	1	20,670 ± 1784	25,480 ± 2002	3072 ± 171	6.87 ± 0.88
	3	20,570 ± 1580	21,410 ± 1710	2636 ± 142	7.02 ± 0.42
	6	15,800 ± 763	15,710 ± 1006	1679 ± 87	6.10 ± 0.55
F13	1	20,630 ± 1560	22,480 ± 1637	2581 ± 133	6.55 ± 0.34
	3	19,853 ± 1020	9789 ± 1059	1123 ± 60	6.55 ± 0.36
	6	12,590 ± 752	12,500 ± 1081	1448 ± 74	6.61 ± 0.35
F17	1	21,800 ± 1213	21,680 ± 1408	2276 ± 121	5.99 ± 0.50
	3	18,020 ± 1280	17,930 ± 1402	1813 ± 88	5.77 ± 0.33
	6	11,300 ± 1100	11,220 ± 994	1304 ± 64	6.62 ± 0.46

## Data Availability

The data presented in this study are available in article and Appendix A.

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
