# Peer review of "Matrix Tablets Based on Chitosan–Carrageenan Polyelectrolyte Complex: Unique Matrices for Drug Targeting in the Intestine"

_pharmaceuticals, 2022, doi:10.3390/ph15080980_

Round 1

Reviewer 1 Report

The manuscript deals with the polymeric system based on polyelectrolyte polymeric complex for prolonged release of two model active pharmaceutical ingredients (APIs). The topic of manuscript is interesting but its scientific quality should be significantly improved before publication in such high ranked journal.

Some suggestions for authors for improvement of manuscript’s quality are given below:

1.       Abstract – line 21: mucoadhesive properties were teste only indirectly, so results can not be directly translated to “in vivo” mucoadhesion!

2.       Line 22 – I suggest rewriting of the last part of the sentence starting: the particularly…..

3.       Line 24 – replace term “enterosolvent” with e.g.enteric coatings, but be careful with using these terms – authors should clearly explain what they meant with this term.

4.       Lines 31 and 32: not all natural polymer are biodegradable!

5.       Line 111: erase yellow mark at the end of the sentence.

6.       Ch. 2.1: authors should explain why they used both types of binders – specially Ca sulphate is not an usual one.

7.       Line 134: replace term “moments” with “intervals”.

8.       Line 186: replace “fixating” with “fixing”.

9.       Line 255: The explanation of selection of proposed formulations should be given.

10.   Fig. 2 D: DSC curve for pure SA should be added to chart and appropriately modify the discussion on DSC results – pages 18-19.

11.   Line 348: add to the text more detailed explanation of the discussed thermal decomposition of the CRG.

12.   Line 349: Authors should supplement explanation how the in the text mentioned “transformation” influence thermal decomposition of CRG.

13.   Line 352: the meaning of the “instantaneous conversion” should be given in the clear and understantable way.

14.   Page 19: Authors should perform interaction study between CRG and CHTS polymer on binary molecular level and translate this data onto the properties of tablets based on both polymers.

15.   Line 373: A more clear meaning of “CRG dissociation” should be discussed – how they connect polymer dissociation with interaction with CHTS!?

16.   Page 20, page 30 and ch. 3.3: Authors wrote that Ca sulphate can partially dissolve and liberate Ca ions which can interact either with SA or –SO3H groups of CRG to form corresponding salts what in turn can effect drug release from the matrix. A comprehensive discussion with supporting experimental data is missing.

17.   Line 388: …. Reported in reference (29).

18.   Line 406: authors should consistently use only one term for API/drug/active substance throughout the whole text.

19.   Lines 408-409: The last sentence of the paragraph is not clear.

20.   Page 29: Authors should experimentally verfy the possibility of ionic interaction of TH and CRG and rewrite corresponding discussion. Also SA can enter into ionic interaction with chitosane (CHTS).

21.   Lines 576-579: a clear explanation why proposed formulations should have mucoadhesive properties should be given.

22.   Page 35: connect rheological results/data with properties of test tablets.

Reviewer 2 Report

The present study, by Alena Komersová et al, is focuses on the more detailed characterization of the chitosan-carrageenan (CRG-CHTS) systems with respect to their potential application for the drug targeting to the intestine.

In brief, the authors found that the CRG-CHTS polyelectrolyte complex essentially replaces the acidoresistant tablet coating. The  complex prevents unintended drug release in the stomach, and subsequently mucoadhesive gel matrix tablet provides  sustained drug release in the small intestine. Thus, it can be argued that the tablets with the chitosan-carrageenan polyelectrolyte complex are promising matrix systems for the treatment of non-specific intestinal inflammations. The methodologies used are the appropriate for this type of studies and the results well documented. Overall, the manuscript is of interest to both the cognizant and non-cognizant readers. 

Reviewer 3 Report

Dear authors,

Presented work is coplexed and a lot of effort could be seen.

There is several questions that should be adressed

- tablet mass preparation should be more clear - legends in table for abrvation shoild be put and why smcc  and caso4 are chosen as fillers

Why only weibul model was choosen and not corelation to more models

Uv vis should be refernced

Dissolution concept of pulling tablets for measurment is not clear

Why are selected f1 11 13 and 17

All results should on figures have SD like fig 2 and 6

Fig 1 picture have no metodology

Fig 3 not clear crossing between formulation...again not clear why only these four formulation are selecyed

Best of lack
